# DNA-Binding Proteins and Passenger Proteins in Plasma DNA–Protein Complexes: Imprint of Parental Cells or Key Mediators of Carcinogenesis Processes?

**DOI:** 10.3390/ijms25105165

**Published:** 2024-05-09

**Authors:** Oleg Tutanov, Aleksei Shefer, Evgenii Shefer, Pavel Ruzankin, Yuri Tsentalovich, Svetlana Tamkovich

**Affiliations:** 1Department of Medicine, Vanderbilt University Medical Center, Nashville, TN 37203, USA; ostutanov@gmail.com; 2Laboratory of Molecular Medicine, Institute of Chemical Biology and Fundamental Medicine, Siberian Branch of the Russian Academy of Sciences, 630090 Novosibirsk, Russia; a.shefer@g.nsu.ru; 3Novosibirsk State University, 630090 Novosibirsk, Russia; 4Laboratory of Applied Inverse Problems, Sobolev Institute of Mathematics, Siberian Branch of the Russian Academy of Sciences, 630090 Novosibirsk, Russia; 5Laboratory of Proteomics and Metabolomics, International Tomography Center, Siberian Branch of Russian Academy of Sciences, 630090 Novosibirsk, Russia; yura@tomo.nsc.ru

**Keywords:** plasma DNA, nucleoprotein complexes, MALDI-TOF mass spectrometry, bioinformatics analysis, DNA-binding motifs, cancer proteins, breast cancer

## Abstract

Knowledge of the composition of proteins that interact with plasma DNA will provide a better understanding of the homeostasis of circulating nucleic acids and the various modes of interaction with target cells, which may be useful in the development of gene targeted therapy approaches. The goal of the present study is to shed light on the composition and architecture of histone-containing nucleoprotein complexes (NPCs) from the blood plasma of healthy females (HFs) and breast cancer patients (BCPs) and to explore the relationship of proteins with crucial steps of tumor progression: epithelial–mesenchymal transition (EMT), cell proliferation, invasion, cell migration, stimulation of angiogenesis, and immune response. MALDI-TOF mass spectrometric analysis of NPCs isolated from blood samples using affine chromatography was performed. Bioinformatics analysis showed that the shares of DNA-binding proteins in the compositions of NPCs in normal and cancer patients are comparable and amount to 40% and 33%, respectively; in total, we identified 38 types of DNA-binding motifs. Functional enrichment analysis using FunRich 3.13 showed that, in BCP blood, the share of DNA-binding proteins involved in nucleic acid metabolism increased, while the proportion of proteins involved in intercellular communication and signal transduction decreased. The representation of NPC passenger proteins in breast cancer also changes: the proportion of proteins involved in transport increases and the share of proteins involved in energy biological pathways decreases. Moreover, in the HF blood, proteins involved in the processes of apoptosis were more represented in the composition of NPCs and in the BCP blood—in the processes of active secretion. For the first time, bioinformatics approaches were used to visualize the architecture of circulating NPCs in the blood and to show that breast cancer has an increased representation of passenger proteins involved in EMT, cell proliferation, invasion, cell migration, and immune response. Using breast cancer protein data from the Human Protein Atlas (HPA) and DEPC, we found that 86% of NPC proteins in the blood of BCPs were not previously annotated in these databases. The obtained data may indirectly indicate directed protein sorting in NPCs, which, along with extracellular vesicles, can not only be diagnostically significant molecules for liquid biopsy, but can also carry out the directed transfer of genetic material from donor cells to recipient cells.

## 1. Introduction

Today, cell-free DNAs are considered molecules with wide clinical applications (diagnosis, assessment of therapy effectiveness, disease prognosis) in various pathologies, including cancer [1,2,3,4]. Cell-free DNA is protected from hydrolysis by endogenous nucleases by packaging into membrane structures (such as apoptotic bodies [5,6,7]) and by binding to proteins [8,9,10], including histones [11]. It should be noted that the resulting nucleoprotein complexes (NPCs) can either circulate freely [12,13] or bind to the surface of blood cells [14,15] and exosomes [16] by interacting with protein [17,18] and DNA [19,20] receptors. At the same time, it is still not entirely clear what cellular processes (apoptosis [21,22,23], necrosis [23,24,25,26], active secretion [23,25,26,27], NETosis [23,28]) lead to the entry of cell-free DNA into the bloodstream, and the contributions of these processes to the generation of extracellular DNA in normal and pathological conditions. Obviously, knowledge of the protein composition of circulating NPCs will provide a better understanding of the homeostasis of cell-free DNA, its biological role in the blood, and, possibly, what mechanisms will be triggered in target cells after the transfer of extracellular DNA into them [29,30,31]. Earlier, we have shown that the NPCs from HF plasma samples contained shorter DNA fragments (~180 bp) than BCP NPCs. However, the share of DNA in the NPCs from cfDNA in blood plasma in HFs and BCPs did not differ significantly, nor did the share of NPC protein from blood plasma total protein. Moreover, bioinformatic analysis after identification of proteins by MALDI-TOF mass spectrometry revealed that, in the presence of a malignant tumor, the proportion of proteins involved in ion channels, protein binding, transport, and signal transduction increased in the composition of blood-circulating NPCs [32]. Since the composition of NPCs can include not only DNA-binding proteins, but also proteins that interact with such proteins but do not bind DNA (passenger proteins), the questions about the significance of those proteins in blood NPCs remains open. Are DNA-binding proteins in NPCs universal (present in normal and pathological conditions)? Is the protein composition of NPCs a footprint of parental cells or, as with sorting into exosomes, is there a directed assembly of proteins into NPCs?

To answer these questions, we isolated native NPCs from the venous blood using affine chromatography. The comparison groups were healthy females (HFs) and primary luminal breast cancer patients (BCPs)—the most common cancer among women. Proteins were identified by MALDI-TOF mass spectrometry and then bioinformatics analysis, and mathematics modeling was used to describe the phenomenon of circulating DNA–protein complexes in normal conditions and in cancerous conditions.

## 2. Results

### 2.1. Concentration of Plasma DNA and DNA from NPCs in the Plasma of HFs and BCPs

The DNA concentration isolated from plasma or NPCs was estimated by qPCR, specific for LINE-1. A significant increase in the plasma DNA concentration was found for untreated BCPs (*n* = 20) compared to HFs (*n* = 15) (median 46 versus 4 ng/mL of blood, *p* = 0.0026, Mann–Whitney U test) (Figure 1a), which coincides with previous studies [15,33]. Moreover, significant differences were found for NPC–DNA in BCPs and HFs (median of DNA was 9.9 versus 1.05 ng/mL of blood, *p* = 0.0041, Mann–Whitney U test) (Figure 1b). Thus, using antibodies against histones by affine chromatography, approximately one-quarter of cell-free DNA can be isolated.

### 2.2. Analysis of DNA-Binding Proteins and Passenger Proteins in Blood NPC Content

Proteins within circulating NPCs were identified by MALDI-TOF mass spectrometry after separation in 10% PAGE and trypsinolysis of proteins in individual gel fragments. Each sample of NPCs was applied to the gel in 5 repeats; a score of 56 and the presence of at least 3 peptides from the protein sequence allowed the identification with high confidence (*p* < 0.05) of 181 and 173 proteins in blood NPCs of HFs (*n* = 15) and untreated BCPs (*n* = 20, Table 1), respectively (Appendix A). It should be noted that 17 out of 43 universal proteins of NPCs were detected in three-quarters of the samples (Appendix A).

To identify DNA and nucleotide-binding proteins within NPCs, the identified proteins were analyzed using QuickGO 2.0 by GO nucleic acid binding (GO:0003676) and nucleotide binding (GO:0000166). We found that 118 proteins (37%) were associated with nucleic acid and nucleotide binding (NA-binding), of which 20 were universal (47% of all universal NPC proteins). The share of NA-binding proteins in the composition of NPC proteins in the blood of HFs amounted to 41%; in the composition of NPC proteins in the blood of BCPs, it was 36% (Figure 2).

To identify the DNA-binding domains in the identified DNA-binding proteins within blood NPCs, the proteomes were analyzed using the Interpro web platform and the Interpro (https://www.ebi.ac.uk/interpro/, accessed on 20 March 2024), PROSITE, Pfam (http://pfam.xfam.org/, accessed on 20 March 2024), SMART (http://smart.embl-heidelberg.de/, accessed on 20 March 2024), and CDD (https://www.ncbi.nlm.nih.gov/Structure/cdd/cdd.shtml, accessed on 20 March 2024) databases (Table 1). Bioinformatics analysis identified 38 types of DNA-binding motifs, the most represented among which are zinc fingers (C2H2, CCCH, PHD, C4, CXXC, and RING types), RNA recognition motif, Homeobox domains, KRAB, loop–turn–loop, and leucine zippers.

In the blood of HF patients, 17 unique DNA-binding motifs were found as part of NPC proteins, and 9 unique DNA-binding motifs were found in the blood of BCPs.

Based on the database of pairwise protein interactions “Human Integrated Protein–Protein Interaction rEference” (HIPPIE version 2.3) [34], modeling of the architecture of possible NPCs was performed using the analysis of proteins detected in the blood of HFs and BCPs. Possible chains were modeled by starting with a DNA-binding protein, followed by one or more passenger proteins. Only “linear” structures were accounted for, where each subsequent protein is attached to the previous one; that is, the first passenger protein is attached to the DNA-binding protein, the second passenger protein is attached to the first passenger protein, and so on. As a result, 23 models of possible NPCs circulating in the blood of healthy subjects and 25 models of possible NPCs circulating in the blood of patients with breast cancer were constructed (Table 2). It is to be noted that a modeled chain of, say, four proteins implies the possibility of existence of the corresponding subchains of two and of three proteins, starting with the DNA-binding protein, e.g., the sequence ALBU–NECD–RL5–M3K14 implies possible existence of the subsequences ALBU–NECD–RL5 and ALBU–NECD. Thus, only the chains of maximal lengths are reported, and 23 and 25 are the numbers of chains of maximal lengths.

### 2.3. Comparative Proteomic Analysis of Circulating NPCs in the Blood of HFs and BCPs

To characterize DNA-binding proteins and passengers proteins identified in blood-circulating NPCs, a bioinformatics analysis was performed using InterPro and InterProScan databases versions 5.15-58 and 5.15-54,24,25, allowing us to identify the GO categories for NPC proteins from HFs (Appendix A) and BCPs (Appendix A) (isoforms not shown). To avoid loss of information and to fully account for the data obtained, all identified proteins were included in the analysis (even if a protein occurred in only one sample). The resulting lists of GO terms for 36 cellular components and 106 biological processes are provided in Appendix A. For 12 proteins, the GO terms were not determined for both categories (cellular components and biological process) (Appendix A).

A comparative analysis of functional enrichment using FunRich 3.13 showed that nuclear (61 and 69%, respectively) and cytoplasmic proteins (45 and 48% each, respectively) were predominant among the DNA-binding proteins of NPCs circulating in the blood of HFs and BCPs (Figure 3a,b).

In addition to DNA-binding proteins, NPCs include proteins that do not directly bind DNA (passenger proteins). The passenger protein analysis is no less important for understanding the processes of NPC formation and its structure, as well as its functions and circulation peculiarities. The comparative analysis of functional enrichment, performed using FunRich 3.13, showed that proteins of cytoplasm (35% each), plasma membrane (27% and 28%), and nucleus (16% and 11%) prevailed among the passenger proteins in the composition of NPCs circulating in the blood of HFs and BCPs (Figure 3c,d).

At the same time, the share of DNA-binding proteins involved in the regulation of nucleotide, nucleoside, and nucleic acid metabolism increases in the BCP blood (43% in cancer and 34% in normal), and the share of DNA-binding proteins involved in the processes of intercellular communication and signal transduction decreases (25% and 27%, respectively, in normal, and 17% in breast cancer) (Figure 4a,b).

Proteins regulating intercellular communication (20% and 22%, respectively) and signal transduction (23% and 24%, respectively) were comparably represented among the passenger proteins of blood NPCs in normal and cancer patients (Figure 4c,d); the representation of passenger protein-mediating energy pathways was three times higher in HFs and BCPs (16% and 5%, respectively), while proteins involved in transport processes were lower (7% and 12%, respectively).

Thus, the composition of NPCs includes both DNA-binding proteins, predominantly of nuclear and cytoplasmic origin, and passenger proteins of cytoplasmic and nuclear as well as plasma membrane origin. It was found that DNA-binding proteins are mainly involved in the processes of the regulation of nucleotide, nucleoside, and nucleic acid metabolism, and passenger proteins are involved in the processes of intercellular communication and signal transduction; an increase in the share of passenger proteins in the BCP blood NPCs involved in unknown biological processes are also noteworthy. Passenger proteins appear to bind to the complexes via DNA-binding proteins and play important roles in transport and protection of cell-free DNA from hydrolysis by nucleases, recognition of the complex by the immune system, internalization, and clearance of NPCs.

### 2.4. Role of DNA-Binding Proteins and Passenger Proteins of NPCs in Breast Cancer Dissemination

Literature analysis and QuickGO 2.0 annotation revealed an involvement of both normal and cancer NPC proteins in such processes as EMT, cell proliferation, invasion, cell migration, angiogenesis, and immune response (Figure 5, Appendix A). However, after separating NPC proteins from DNA-binding and passenger proteins, it was found that passenger proteins are the main participants in these processes.

Comparative analysis of NPC proteins circulating in the blood of HFs and BCPs by individual processes revealed the following regularities:-EMT inhibitory proteins are absent in NPCs; however, more EMT-stimulated passenger proteins were detected in cancer than in normal (5 vs. 1, Figure 5a);-No proteins regulating the development of vasculogenesis were found in the DNA-binding proteins of NPCs from BCP blood, whereas they were comparably represented in passenger proteins at normal and breast cancer (Figure 5b);-Proteins involved in the regulation of cell proliferation are more frequently detected, both in the composition of DNA-binding proteins and in the composition of passenger proteins of blood NPCs of BCPs, with more protein inhibitors of this process detected in the composition of NPC passenger proteins in HF blood (8 vs. 1, Figure 5c);-Proteins that inhibit cell migration are comparably represented in the NPCs of blood of HFs and BCPs, while more proteins stimulating this process were detected in the composition of passenger proteins in pathology (22 vs. 14, Figure 5d);-Invasion-inhibitory proteins are absent in NPCs; however, more of them were detected in cancer than normal patients (19 vs. 11, Figure 5e);-More proteins involved in the regulation of immune response were detected in the composition of NPC passenger proteins in the blood of BCPs than in HFs (14 vs. 8, Figure 5f).

Thus, while the representation of DNA-binding proteins is comparable in normal and breast cancer samples, the representation of passenger proteins, which are involved in cell proliferation, invasion, cell migration, and immune response, is increased in pathology. In addition, the repertoire of proteins in the composition of NPCs is fundamentally altered (Appendix A). The obtained results indirectly indicate that only cell-free DNA in complex with proteins possesses biological activity in both normal and pathology.

### 2.5. NPC Proteins Reflect the Origin of Cell-Free DNA

On the next stage, to identify proteins involved in the mechanisms of cell-free DNA appearance in the bloodstream (apoptosis, necrosis, active secretion), the proteomes of NPCs from the blood of HFs and BCPs were analyzed using QuickGO 2.0 using GO:00006915 (apoptotic process, GO:00006915), GO:0070266 (necrotic process, GO:0070266), and GO:0046903 (secretion, GO:0046903).

It was shown that six (RALB, RPS6KB1, OXTR, HOXAI3, TRAF3, MAP3K5) and three (NAIF1, BIRC5, CASP1) unique proteins in NPCs are associated with apoptosis in blood of HFs and BCPs, respectively (Figure 6). In addition, one necrotic-related protein (MLKL) and three secretion-related proteins (SYT3, RALA, SLC4A5) were detected in HF NPCs, indirectly indicating that normal cell-free DNA appears in the blood mainly as a result of programmed cell death. Proteins involved in necrotic processes were not detected in the composition of NPCs of the BCP blood; however, five proteins were associated with active secretion (RALA, SLC4A5, VEGFA, SYT10, NEUROD1). These data indirectly indicate the predominance of secreted DNA over apoptotic DNA in the blood of cancer patients (Figure 6).

Of the 16 proteins in the composition of NPCs associated with apoptosis, necrosis, or active secretion, only 2 proteins were DNA-binding (HOXAI3 and NEUROD1), which, once again, indicates the high significance of passenger proteins in DNA–protein complexes.

### 2.6. NPC Proteins Are Not an Imprint of Parental Cells

To determine whether the proteins we identified were found in breast neoplasms, the list of proteins was analyzed, using FunRich 3.13 software to search publicly available databases for proteins in breast cancer—HPA and DEPC. Only three breast cancer-associated proteins from NPCs circulating in the blood of BCPs were annotated in the HPA database (Figure 7).

Furthermore, a search for proteins identified within NPCs in the DEPC database revealed only 23 proteins that were previously recognized as associated with breast cancer. Thus, 84% of the proteins identified in the blood NPCs of BCPs in this study were not previously annotated in the databases of proteins differentially expressed in tissues of cancer patients. The obtained data may indirectly indicate the directed sorting of proteins into NPCs, which, along with extracellular vesicles, can not only be diagnostically significant molecules for liquid biopsy, but can also carry out the directed transfer of genetic material from donor cells to recipient cells.

## 3. Discussion

In recent years, there has been increasing evidence that complexes of DNA associated with proteins and lipids are more effective than naked DNA in gene delivery to the nucleus [35]. This phenomenon suggests that proteins included in NPCs are not only protectors of DNA from DNAases and phosphodiesterases [36] in blood, but also tags providing targeted delivery of genetic material to target cells, and some of them (transcription factors, enzymes, etc.) are capable of triggering various processes affecting the further fate of cells.

The majority of researchers agree that more than 99% of DNA in the blood is of endogenous origin [1,4,6,7], and its concentration increases under pathological conditions (trauma [37], autoimmune diseases [38], cancer [3,4,15,31,33], etc.). In independent studies of DNA kinetic analysis, size and ends profiling [25,39,40,41], strong evidence has been obtained that part of the cell-free DNA is a product of active cellular secretion. Our results on the protein composition of NPC that were analyzed by STRING 12.0 software after identification of proteins indirectly confirm the data of these authors: in BCP blood, the representation of apoptosis proteins decreases and the representation of proteins involved in secretion processes increases [32]. The absence of necrosis-associated proteins in NPC in the blood of BCPs at stage I of the disease seems logical, since necrotic processes in malignant pathologies are characteristic of the terminal stages, when severe ischemia of tumor tissues is observed. In the following study, we showed, on smaller but more homogeneous groups of HFs and BCPs using the Interpro web platform, PROSITE, and Pfam databases, that the NPC protein cargo from HF blood was enriched with proteins involved in the negative regulation of cell proliferation, and, in BCP blood, proteins involved in EMT, invasion, and cell migration were observed [42].

The fact that affine chromatography on a sorbent with immobilized polyclonal anti-histone antibodies in both normal and cancer allows the isolation of only a quarter of plasma DNA indicates that, in addition to DNA fragments that are multiples of the nucleosome, DNA is present in the blood as a part of apoptotic bodies, as well as in the form of higher molecular weight DNA fragments that cannot be isolated using the approach used.

Using bioinformatics analysis, 38 types of DNA-binding motifs were identified in NPC proteins in this study, which allowed us to conditionally divide NPC proteins into nucleic acid-binding and passenger proteins, which account for 62% of all proteins. The share of proteins regulating nucleic acid metabolism increases in the composition of DNA-binding proteins and the share of proteins involved in the processes of intercellular communication and signal transduction decreases, while the share of transport proteins increases in the composition of passenger proteins and the share of proteins of energetic biological pathways decreases.

To elucidate the molecular regulatory roles underlying blood NPC-mediated tumor progression, a bioinformatics analysis of DNA-binding proteins and passenger proteins was performed. Currently, there is practically no information on the composition of DNA–protein complexes circulating in the blood, especially about their architecture. Technical difficulties in identifying blood NPC proteins are associated, first of all, with the difficulty of obtaining native complexes without admixtures of blood plasma proteins that are not part of such complexes. Nevertheless, we found that the representation of passenger proteins in NPCs that are involved with EMT, cell proliferation, invasion, cell migration, and immune response increases in cancer.

Thus, the identification of proteins and the establishment of NPC architecture are of fundamental importance for understanding the molecular mechanisms of the processes that ensure the transfer of genetic information and signals between cells. Establishment of the functional role of DNA-binding proteins and passenger proteins significantly adds to the picture of biogenesis and functional role of circulating NPCs in cancer development. In addition, the identified DNA-binding motifs unique to tumor-specific proteins can be used to enrich cell-free tumor DNA, which can improve the efficiency of cancer diagnosis by “liquid biopsy” and create fundamental prerequisites for possible optimization of antitumor therapy.

## 4. Materials and Methods

### 4.1. Patients

Blood samples from HFs (*n* = 15, mean age 48 ± 2.1 years) were obtained from the Medical Scientific and Educational Center of the V. Zelman Institute of Medicine and Psychology, Novosibirsk State University. The donor group was formed on the basis of a questionnaire as well as a clinical examination. All women underwent an ultrasound examination of the breast and pelvic organs, mammography, low-dose computed tomography of the lungs, and general and biochemical blood tests. Women with reproductive system disorders, endocrine and metabolic factors, and the presence of genetic and exogenous factors were excluded from the study.

Blood samples from untreated BCPs (*n* = 20, mean age 53 ± 2.9 years) were obtained from the Novosibirsk Regional Clinical Oncology Dispensary. The clinicopathological parameters of BCPs are presented in Table 3.

Expression of estrogen (ER) and progesterone (PR) receptors and HER-2 status were determined by the immunohistochemical examination of tissue samples after surgery, as described [43].

### 4.2. The Isolation of Histone-Containing NPC from Blood Plasma

Blood plasma histone-containing NPCs were isolated using affine chromatography, as previously described [32]. Briefly, venous blood (9 mL) was collected in K3EDTA spray-coated vacutainers (Improvacuter, China, cat. No. 694091210), immediately mixed using a rotary mixer, placed at +4 °C, and processed within 1 h after taking the blood. The blood cells were pelleted by centrifugation for 20 min at 290× *g* and 4 °C, and then plasma-centrifuged a second time at 1200× *g* for 20 min. Plasma samples were aliquoted and stored at −80 °C. The aliquots were thawed once before use.

Affine sorbent with immobilized anti-histone antibodies (the rabbit polyclonal anti-H2A; (PAQ850Hu01), anti-H2B; (PAQ006Hu01), and anti-H3 (PAA285Mi01) antibodies (Cloud-Clone Corp. (Wuhan, Hubei, China)) was synthesized from bromocyan-activated Sepharose, as previously described [41]. For the isolation of histone-containing NPC by affine chromatography, plasma (0.8 mL) was loaded onto the sorbent that contained 3 mg total immunoglobulins and 3 g CL-4B Sepharose, and incubated for 1 h at 4 °C. Then, the sample was loaded onto affine sorbent again. After washing the column with PBS containing 5 mM EDTA, PBS containing 5 mM EDTA and 0.05% Tween-20, and again with PBS containing 5 mM EDTA, NPCs were eluted from the column in the opposite direction with glycine buffer and neutralized with borate buffer. The NPC samples were concentrated on Centricon 3 kDa filters for 4 h at 4000× *g*, +4 °C.

### 4.3. Characterization of Nucleic and Protein Components of NPCs

The DNA from plasma and from histone-containing NPCs was isolated using the “DNA Isolation Kit” (BioSilica Ltd., Novosibirsk, Russia) according to the manufacturer’s protocols and concentrated by precipitation with trimethylamine and glycogen, as described earlier [15]. The concentration of DNA was measured by quantitative polymerase chain reaction, specific for long interspersed nuclear element 1 (LINE-1) repetitive elements, as described earlier [15]. Genomic DNA from human leukocytes served as a standard for obtaining the calibration curves.

Individual plasma NPC samples were separated according to their molecular weight using 10% SDS disc-electrophoresis and identified by mass spectrometry, as described earlier [42]. Mass spectra were registered at the Center of Collective Use “Mass spectrometric investigations” SB RAS on an Ultraflex III MALDI-TOF/TOF mass spectrometer (BrukerDaltonics, Bremen, Germany) in positive mode, with the range 700–3000 Da, and with 2,5-dihydroxybenzoic acid as a matrix. Proteins were identified by searching for appropriate candidates in annotated NCBI and SwissProt databases using Mascot 2.6.1 software (Matrix Science Ltd., London, UK, www.matrixscience.com/search_form_select.html, accessed on 10 May 2023). The following parameters were used for searches: acceptable mass deviation of the charged peptide (50 ppm)—0.05 Da; acceptable number of missed cleavage sites—2; carbamidomethylation of cysteine residues was chosen as a fixed modification and the presence of oxidized methionine residues was chosen as a variable modification; and identification reliability was not lower than 95%.

### 4.4. Bioinformatics and Gene Ontology (GO) Analysis of NPC Proteins

The presence of DNA-binding motifs within proteins was analyzed using the Interpro web platform and Interpro (https://www.ebi.ac.uk/interpro/, accessed on 8 January 2024), PROSITE, Pfam (http://pfam.xfam.org/, accessed on 10 January 2024), SMART (http://smart.embl-heidelberg.de/, accessed on 12 January 2024), and CDD (https://www.ncbi.nlm.nih.gov/Structure/cdd/cdd.shtml, accessed on 15 January 2024) databases. GO profiling of NPC proteins involved in the cell migration and motility, immune response, vasculature development, and cell proliferation was performed using QuickGO 2.0 annotation terms (lists of obtained proteins were searched against GO terms: cell motility (GO:0048870), cell migration (GO:0016477), negative regulation of cell motility (GO:2000146), immune response (GO:0006955), negative regulation of immune response (GO:0050777), vasculature development (GO:0001944) negative regulation of vasculature development (GO:1901343), cell population proliferation (GO:0008283), and negative regulation of cell population proliferation (GO:0008285)) [44,45,46]. The involvement of NPC proteins in cancer invasion and EMT was routinely analyzed by searching the PubMed database for relevant publications for each protein.

The search for cancer prognostic proteins in the NPC proteome of BCPs was conducted using the HPA (http://www.proteinatlas.org/, accessed on 31 May 2023) and dbDEPC 3.0 (accessed on 31 May 2023) databases for breast cancer.

Modeling of the architecture of possible NPCs was performed using the analysis of proteins detected in the females’ blood and database of pairwise protein interactions “Human Integrated Protein–Protein Interaction rEference” (HIPPIE version 2.3) [34].

### 4.5. Statistical Analysis

Statistical calculations were performed using Statistica 6.0 software. All data were expressed either as medians with interquartile ranges or as means with standard errors. To evaluate the differences, the Mann–Whitney U test was performed. *p* < 0.05 was considered statistically significant.

## Figures and Tables

**Figure 1 ijms-25-05165-f001:**
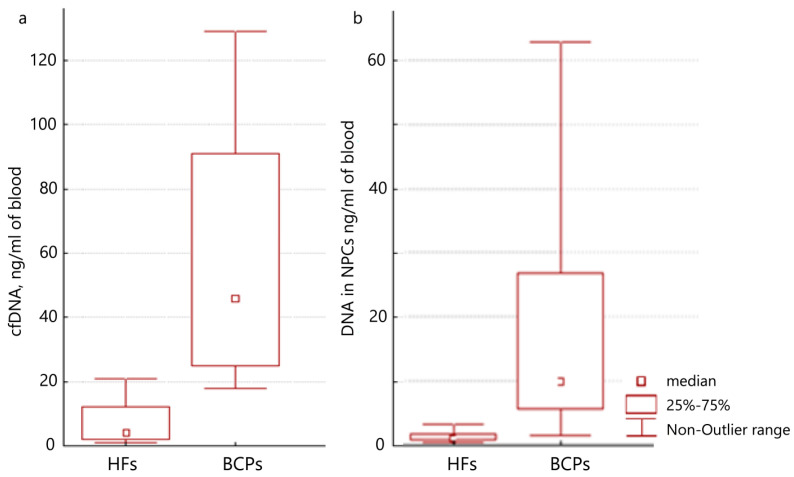
Quantification of DNA from plasma or from NPCs in the plasma of HFs and BCPs. (**a**) cell-free DNA concentration; (**b**) NPC–DNA concentration. Tukey box plots of DNA. Median DNA concentration with 25–75% and non-outlier range bars indicated.

**Figure 2 ijms-25-05165-f002:**
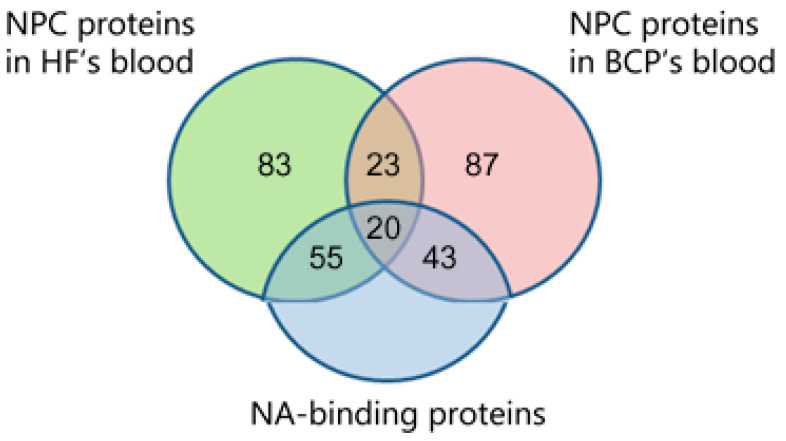
Venn–Euler diagram of DNA-binding proteins in NPCs from HF and BCP blood; composed using QuickGO 2.0 and FunRich 3.13 software.

**Figure 3 ijms-25-05165-f003:**
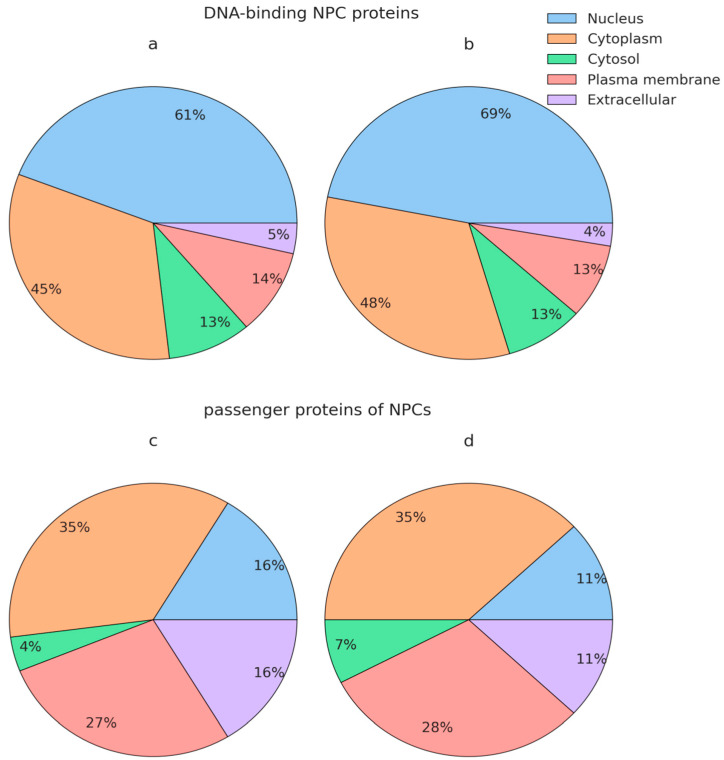
GO analysis of NPC proteins by cellular components. (**a**,**c**) NPC proteins of HFs and (**b**,**d**) NPC proteins of BCPs.

**Figure 4 ijms-25-05165-f004:**
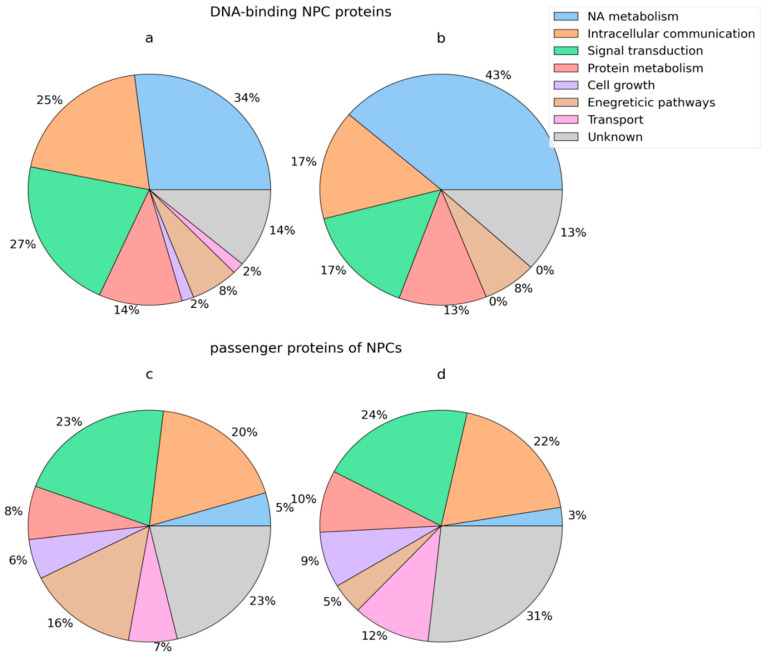
GO analysis of NPC proteins by biological processes. (**a**,**c**) NPC proteins of HFs and (**b**,**d**) NPC proteins of BCPs.

**Figure 5 ijms-25-05165-f005:**
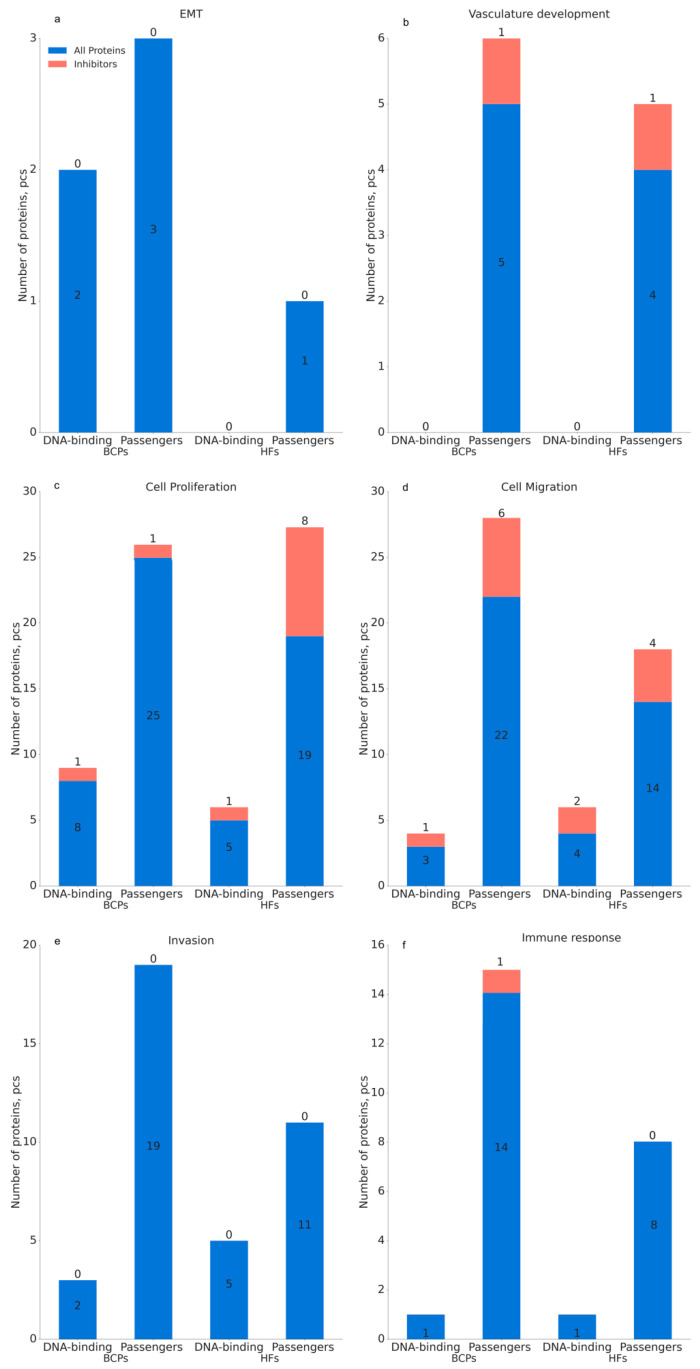
NPC proteins associated with carcinogenesis in blood of HFs and BCPs: (**a**) EMT, (**b**) vasculature development, (**c**) cell proliferation, (**d**) cell migration, (**e**) invasion, (**f**) immune response.

**Figure 6 ijms-25-05165-f006:**
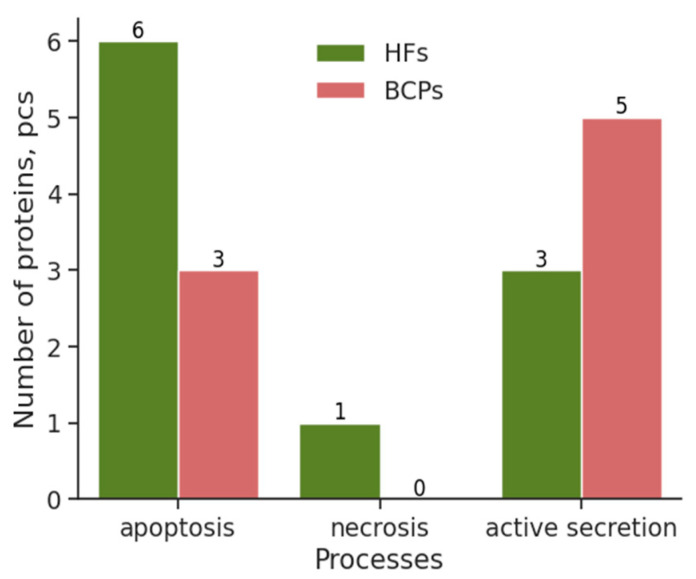
Proteins involved in apoptosis, necrosis, and active secretion as part of NPCs circulating in the blood of HFs and BCPs.

**Figure 7 ijms-25-05165-f007:**
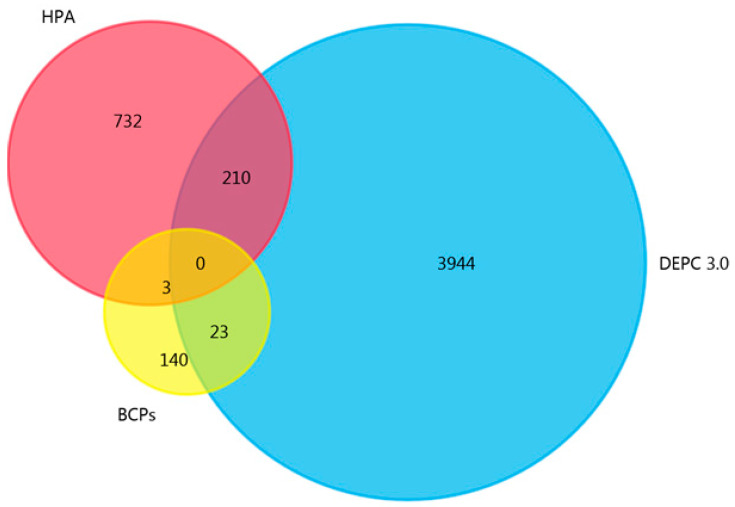
Venn–Euler diagram of NPC proteins from BCP blood in dbDEPC and dbHPA, composed using QuickGO 2.0 and FunRich 3.13 software.

**Table 1 ijms-25-05165-t001:** NA-binding motifs in the blood NPC proteins of HFs and BCPs.

DNA-Binding Domains	Proteins of NPCs (Gene Name)		
	Universal Proteins	Unique Proteins of HFs	Unique Proteins of BCPs
Zinc finger C2H2-type	UBP22	KLF10 SLC2A4RG ZNF479	EGR4 GAS2L3 PLAGL2 RNF222 ZNF461 ZNF75A ZNF75CP ZNF75D
RNA recognition motif	U2AF1	ESRP1 MTHFSD SNRNP35 SREK1 SRSF5 ZRSR2P1/ZRSR1	ENOX2 MYEF2 REXO5
Homeobox domain	HOXC5	HOXA13 HOXB4 HOXC8	CDX2 HOXB9 POU5F1B UNCX
Zinc finger, CCCH-type	U2AF1	MKRN2 TRMT1 ZRSR2P1/ZRSR1	
Basic-leucine zipper (bZIP)	CREM JUND		CHST7 JUN
Jun-like transcription factor	JUND		JUN
Krueppel-associated box (KRAB)	ZNF479		POGK ZNF461 ZNF75A ZNF75CP ZNF75D
Basic helix–loop–helix (bHLH) domain		MSGN1 NEUROD4	HAND1 NEUROD1
Zinc finger, PHD-type		AIRE FBXL19	PHF1 SP110
HSR domain		AIRE	SP110
Nucleic acid-binding, OB-fold		MCM3	NABP2
SAND domain		AIRE	SP110
Histone H2a/H2b/H3		H2AJ/H2AFJ	
(Armadillo-type fold) MIF4G-like domain superfamily		EIF5	
Anticodon-binding domain		TARSL2	
DNA/RNA-binding repeats in PUR-alpha/beta/gamma		PURA	
F-box domain		FBXL19	
G-patch domain		GPATCH4	
High mobility group box domain		HMGB4	
Interferon regulatory factor DNA-binding domain		IRF2	
MCM domain		MCM3	
Neuronal helix–loop–helix transcription factor		NEUROD4	
PIN domain		FCF1	
TATA box-binding protein-associated factor RNA polymerase I subunit A-like		TAF1A	
TGS domain		TARSL2	
Transcription initiation factor TFIID subunit 12 domain		TAF12	
Translation initiation factor IF2/IF5, zinc-binding		EIF5	
Zinc finger, CXXC-type		FBXL19	
Zinc finger, RING-type		MKRN2	
Nascent polypeptide-associated complex NAC domain			NACA
Brinker DNA-binding domain			POGK
Bromodomain			SP110
POU domain			POU5F1B
HSPH1, nucleotide-binding domain			HSPH1
HTH CenpB-type DNA-binding domain			POGK
THAP-type zinc finger			THAP7
(Armadillo-type fold) Uncharacterised domain NUC173			RRP12
Zinc finger C4-type			LMCD1 TRIM68

**Table 2 ijms-25-05165-t002:** Possible NPCs in the blood of HFs and BCPs. Symbolic gene names correspond to HIPPIE conventions. The numbers present the Entrez Gene IDs of the genes.

NPCs in the Blood of HFs		NPCs in the Blood of BCPs	
U2AF1–SMD1	7307–6632	JUN–GOGA2–STRN3	3725–2801–29966
UBP22–MDM4	23326–4194	U2AF1–SMD1	7307–6632
FCF1–MAOX	51077–4199	PLAL2–CTSR1–VINEX	5326–117144–10174
U1SBP–RAB6A	11066–5870	GAS2L3–BIRC5	283431–332
U1SBP–SERPH	11066–871	B2R5B3–RM35	3014–51318
B2R5B3–HAT1	3014–8520	B2R5B3–RANG	3014–5905
SRSF5–M3K14–DNJB6	6430–9020 –10049	B2R5B3–BIRC5	3014–332
SRSF5–RM47–RUSD4	6430–57129–84881	H2BC21–UBP12	8349–219333
H2BC21–RHG30	8349–257106	H32–BIRC5	126961–332
H2BC21–PRR12	8349–57479	H4–NOL9	8370–79707
H4–GLYC	8370–6470	H4–SMD1	8370–6632
H4–HAT1	8370–8520	H4–CL043	8370–64897
H4–SMD1	8370–6632	H4–NAIF1	8370–203245
H4–IN80E	8370–283899	H4–SPAT5	8370–166378
H4–PRR12	8370–57479	H4–IN80E	8370–283899
H4–LMNB2	8370–84823	H4–RL5–M3K14	8370–6125–9020
H4–LC7L2	8370–51631	H4–RL5–NECD	8370–6125–4692
H4–FKB11	8370–51303	H4–NECD–RL5–M3K14	8370–4692–6125–9020
H4–INT9	8370–55756	H4–RANG	8370–5905
ALBU–MDM4	213–4194	H4–CSN2	8370–9318
ALBU–THRB	213–2147	H4–TRAM1	8370–23471
ALBU–QTRD1	213–79691	H4–PNISR	8370–25957
ALBU–LC7L2	213–51631	H4–ESYT2–DJC25	8370–57488–548645
		ALBU–VINEX–CTSR1	213–10174–117144
		ALBU–NECD–RL5–M3K14	213–4692–6125–9020

**Table 3 ijms-25-05165-t003:** Clinical characteristics of untreated BCPs.

Clinical Characteristics		N (%)
Tumor stage	T1	20 (100%)
Lymph node status	N0	20 (100%)
Distant metastasis	M0	20 (100%)
Receptor status	ER-positive PR-positive	20 (100%) 20 (100%)
HER-2 status	Negative	20 (100%)
Histologic grade	II III	19 (95%) 1 (5%)
Histological type	Invasive ductal carcinoma	20 (100%)

## Data Availability

The data presented in this study are available on request from the corresponding author.

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
