# Peer review of "DNA-Binding Proteins and Passenger Proteins in Plasma DNA–Protein Complexes: Imprint of Parental Cells or Key Mediators of Carcinogenesis Processes?"

_ijms, 2024, doi:10.3390/ijms25105165_

Round 1

Reviewer 1 Report

Comments and Suggestions for Authors

The manuscript “DNA-binding proteins and passenger proteins of plasma DNA-protein complexes: imprint of parental cells or key mediators of carcinogenesis processes.” discusses the composition and architecture of histone-containing nucleoprotein complexes isolated from the blood plasma of healthy female subjects and breast cancer patients and explores their relationship with crucial steps of tumor progression including epithelial-mesenchymal transition, cell proliferation, invasion, cell migration, angiogenesis, and immune response.

The findings presented in the manuscript are new, valuable, and exciting.

The entire manuscript is correctly written with a sufficiently described Materials and Methods section and an adequately presented Result section complemented with extensive tabular and illustrative data ( manuscript text and supplementary data). The discussion section is appropriate, and a comprehensive and up-to-date reference list accompanies the manuscript.

Acceptance of the manuscript in its current state is suggested.

Author Response

We heartily thank the reviewer for such an appreciation of our work.

Reviewer 2 Report

Comments and Suggestions for Authors

This is an interesting and important study reporting a set of compelling observations on the functionality of the DNA-binding proteins and passenger proteins of plasma DNA-protein complexes. The manuscript is generally well written and concise. Conducted analyses produced coherent and thought-provoking data. The only issue is some misrepresentation of previous studies conducted by the same research group. In fact, although some interesting data were reported by these authors in 2023 (e.g., references 41 and 42), the corresponding information is not mentioned in introduction section and is almost undiscussed in general. The authors are encouraged to add corresponding information to the introduction section and also add discussion of how this current study is different from the previous works.        

41. Tutanov, O.; Shefer, A.; Tsentalovich, Y.; Tamkovich, S. Comparative analysis of molecular functions and biological role of proteins from cell-free DNA-protein complexes circulating in plasma of healthy females and breast cancer patients. Int. J. Mol. Sci. 2023, 24(8), 7279. doi: 10.3390/ijms24087279.

42. Shefer, A.; Tutanov, O.; Belenikin, M.; Tsentalovich, Y.P.; Tamkovich, S. Blood plasma circulating DNA-protein complexes: involvement in carcinogenesis and prospects for liquid biopsy of breast cancer. J. Pers. Med. 2023, 13(12), 1691. doi: 535 10.3390/jpm13121691

Comments on the Quality of English Language

English is generally fine.

Author Response

Dear Reviewer,

we are presenting to you the revised version of our manuscript entitled “DNA-binding proteins and passenger proteins of plasma DNA-protein complexes: imprint of parental cells or key medi-ators of carcinogenesis processes?”, authored by Oleg Tutanov, Aleksei Shefer, et al. On behalf of myself and co-authors, I would like to thank you for helpful suggestions and comments on our manuscript. We have revised the manuscript, fully addressing your comments as described below point by point:

We thank the reviewer for appreciating our study to analyze the architecture of NPCs circulating in the blood in both normal and breast cancer. This manuscript is the final, part of a triptych. Indeed, our earlier studies in this area should have been highlighted more extensively. The first paper (IJMS, 2023) involved a larger and more heterogeneous group of women. In particular, we analyzed proteins within NPCs from the blood of breast cancer patients with the presence of metastases and stages I and II. In the next study (JPM, 2023), we reduced sample sizes, with all patients being stage 1 without metastases. The current manuscript uses primary data (MALDI-TOF Data) on the same women for analysis as the previous paper (JPM, 2023). However, in the current work, a completely different bioinformatics analysis was performed: DNA-binding motifs were identified and NPC models were constructed for the first time in this area of research. We have expanded the Introduction (p. 2) and Discussion (pp. 12-13) by adding data from our past studies (IJMS, 2023 & JPM, 2023); the new text is highlighted in blue.

We hope that we have addressed the reviewer concerns to your satisfaction and the revised manuscript can be considered for publishing in Special Issue "Recent Analysis and Applications of Mass Spectrum in Biochemistry 2.0" of International Journal of Molecular Science.

Best regards,

Reviewer 3 Report

Comments and Suggestions for Authors

This manuscript has presented some interesting screening data of nucleoprotein complexes isolated from the blood of 15 healthy females and 20 females with early stage breast cancer. The extraction methods and bioinformatics analysis are suitable for this type of study and the results are presented in a manner that is easy to follow. The use of various bioinformatics screening programmes is a convenient initial step, and it is noted that no validation of results has been included. I feel that the manuscript is worthy of publication, however the authors could consider the following comments in relation to the interpretation of their data.

1. The data presented are all an average of results obtained from the 15 healthy females and 20 breast cancer patients. Because of the need to clearly depict results, it is not possible to discern whether there was any consistency between the results from individual members of these 2 groups of women or whether the findings represent an average of extensively heterogeneous results from individual members of the 2 groups. As the authors are proposing future use of their data for the development of diagnostic markers or for gene transfer, a more comprehensive description of results, in particular their heterogeneity, would be appropriate.

2. How do the findings obtained in this study relate to results described in the authors’ previous publication (reference 41)? Can any of the results be cross-referenced? In addition, did the 2 studies involve blood samples from the same donors?

Comments on the Quality of English Language

English language is good, however there are minor grammatical errors as well as incorrect naming of scientific terminology (e.g. affine (sic) chromatography) that will require correction.

Author Response

Dear Reviewer,

we are presenting to you the revised version of our manuscript entitled “DNA-binding proteins and passenger proteins of plasma DNA-protein complexes: imprint of parental cells or key medi-ators of carcinogenesis processes?”, authored by Oleg Tutanov, Aleksei Shefer, et al. On behalf of myself and co-authors, I would like to thank you for helpful suggestions and comments on our manuscript. We have revised the manuscript, fully addressing your comments as described below point by point:

  1. When analyzing proteins within NPCs circulating in the plasma of HFs and breast cancer patients, the manuscript, as in most cases, summarizes data from all samples, which is a common trend when presenting such results (Proteomics analysis of plasm exosomes in early pregnancy among normal pregnant women and those with antiphospholipid syndrome // Heliyon. 2024 Apr 12;10(8):e29224. doi: 10.1016/j.heliyon.2024.e29224; Proteomic analysis of exosomal proteins associated with bone healing speed in a rat tibial fracture model // Biomed Chromatogr. 2024 May;38(5):e5846. doi: 10.1002/bmc.5846; Pre-diagnostic plasma proteomics profile for hepatocellular carcinoma // J Natl Cancer Inst. 2024 May 1:djae079. doi: 10.1093/jnci/djae079, etc).To partially address the heterogeneity of samples, we significantly reduced the groups of women in the study. Specifically, we analyzed NPC proteins from the blood of untreated breast cancer patients (ER+PR+HER-2-) without metastases and only at stage I. This analysis revealed patterns that the authors present in Table 1, Figures 3-6. Realizing that the analysis of individual NPCs will be of most interest, we have listed in Table 2 the possible variants of NPCs that may exist in individual samples based on the results of protein identification in individual samples.
  2. This manuscript is the final part of a triptych. Indeed, our earlier studies in this area should have been highlighted more extensively. The first paper (IJMS, 2023) involved a larger and more heterogeneous group of women. In particular, we analyzed proteins within NPCs from the blood of breast cancer patients with the presence of metastases and stages I and II. In the next study (JPM, 2023), we reduced sample sizes, with all patients being stage 1 without metastases. The current manuscript uses primary data (MALDI-TOF Data) on the same women for analysis as the previous paper (JPM, 2023). However, in the current work, a completely different bioinformatics analysis was performed: DNA-binding motifs were identified and NPC models were constructed for the first time in this area of research. We have expanded the Introduction (p. 2) and Discussion (pp. 12-13) by adding data from our past studies (IJMS, 2023 & JPM, 2023); the new text is highlighted in blue.

Misprints have been corrected in the revised version of the manuscript.

We hope that we have addressed the reviewer concerns to your satisfaction and the revised manuscript can be considered for publishing in Special Issue "Recent Analysis and Applications of Mass Spectrum in Biochemistry 2.0" of International Journal of Molecular Science.

Best regards,
